# Development of a Class Model for Improving Creative Collaboration Based on the Online Learning System (Moodle) in Korea

**Eunjoo Kim [1],\* , Hangsik Park [1],\* and JungUn Jang [2]**

[1] The Faculty of Liberal Arts, Eulji University, Seongnam 13135, Korea
[2] Department of Optometry, Eulji University, Seongnam 13135, Korea; jju@eulji.ac.kr
\* Correspondence: kej70@eulji.ac.kr (E.K.); parkhs@eulji.ac.kr (H.P.)

**Abstract:** This study aims to develop a Modular Object-Oriented Dynamic Learning Environment (Moodle)-based online learning system and derive a class model to enhance learners' creative collaboration capabilities. The research method for this is first to analyze the functional elements of the online learning system to enhance creative collaboration among the functions of the Moodle online learning system. Second, a class model is derived to enhance creative collaboration by applying the constructivist teaching–learning principle. Third, we conduct a verification of the effectiveness of the Moodle-based class model that was developed to enhance creative collaboration. The results of the study show that, among the functions of the Moodle online learning system, forms, real-time conferencing, reciprocal evaluation, Wiki, and blogs have the ability to enhance creative collaboration. Second, the class model for promoting creative collaboration based on a mood-based online learning system consists of self-reflections, learner-driven learning, professional learning, practical task, and the role of the processor. Third, after verifying the application effect of the Moodle-based class model for university students, a significant effect is found in the creative collaboration of university students. The results of this study suggest that: First, the Moodle online learning system should be utilized to provide learners with a variety of online learning opportunities to promote creative collaboration. Second, the teaching–learning operation manual needs to be developed later in order for the class model to be effectively implemented at the classroom site to promote creative collaboration based on the Moodle online learning system.

**Keywords:** creative collaboration; online learning system; Moodle

## 1. Introduction

Recently, various online education methods using IT technology have been introduced to the field of education. However, for most online learning, system design and effect studies related to learners' self-directed learning ability are dominant, and research on the educational method for the improvement of creative collaboration is insufficient. Creative collaboration aims to achieve creative—novel and appropriate—outcomes through sharing of creative processes between the cooperatives. It is known that creative collaboration affects creative outcomes through psychological mechanisms such as reflections-in-action, brainstorming, compaction, and interdependency [1]. An educational method to promote this creative collaboration is the Moodle Online Learning System, which is an open-source learning system. Moodle's learning activity tools support learning activities through mutual cooperation among members, and construct knowledge by naturally experiencing the knowledge sharing process that occurs for learners [2,3]. The open-source learning management system, Moodle, is useful for creating an effective online learning community. It also encourages learners to participate in the

learning process and helps professors who aim to improve the learning performance of their students through creative collaboration.

The development of IT technology in the 21st century and the transformation into a knowledge and information society led to many changes, even for the field of education. In particular, E-learning education was expanded with an emphasis on innovation in the educational field, and interest in teaching methods for fostering higher spirits, such as learners' creativity, problem-solving skills, and cooperative skills, was increased through various interactions between professors and learners, and between learners and other learners. In particular, with the shift of the education paradigm to an E-learning environment, learners are perceived as being members of a learning community pursuing common values and goals rather than beneficiaries of the teaching content delivery system. This shift also emphasizes that knowledge and experience—limited to a small number of people in the past—should not be limited to only a small number of people but should be able to be shared anytime and anywhere [4,5].

This shift in the educational paradigm requires a creative collaboration function as one of the necessary education methods for learners. Creative collaboration is essential to achieving that goal, which entails intensive exchange and dialogue among collaborators, control of different opinions and conflicts, and active participation in alternative decisions. If effective collaboration is not obtained, then serious conflicts occur and immature conclusions among collaborators are produced. This outcome negatively affects the creativity of the collaboration results. The study of Beckman et al. (2018) highlights collaborative teamwork through convergence between culture and arts, and business in the classroom [6]. Creative collaboration began in the late 1970s, and the collaborative performance of tasks and knowledge sharing among group members helped to increase creativity within the team. A typical theory of creative collaboration is Osbon's group brainstorming. Group brainstorming can enhance team creativity through the sharing of thoughts and ideas among group members in collaborative learning. This means free exchange of ideas among team members is effective in generating new ideas [7,8].

In order to develop a creative collaboration function for learners in the educational field, it is necessary to have an open approach to knowledge, avoid developing a general approach to that knowledge, and explore other possibilities and alternatives. It is difficult to form this type of productive attitude while conducting a professor-centered one-way lecture class in a traditional classroom. Recognizing this as a problem, in many studies, an online multi-user environment is being considered by some scholars as a tool for creative collaboration as a way to explore and solve educational methods in the online learning environment [9]. In particular, with the development of information and communication, online learning systems are being actively developed in education sites to share information and thinking.

An online learning system is important in an educational environment that can ensure interactivity. Weigel (2002) argued that the core of online learning should be considered from the beginning of the system's development phase so learners can have a voluntary experience on their own [10], which basically involves an autonomous learning environment in which learners are masters, where diverse information is developed, and where learners can share knowledge and information with others, such as peer learners and professors [11]. In addition, online learning systems can promote learners' autonomous learning, problem-solving, and critical thinking skills. The advantages of online learning are that educational institutions have already built and operate their own online learning systems to effectively conduct web-based education. However, each educational institution has various burdens, including the costs of building the system, maintenance costs, and the costs of securing professionals with relevant knowledge and skills. Nevertheless, there is still a growing demand for the use of online learning systems in education sites. Thus, open-source online learning systems have emerged as a new alternative to meeting these needs [4].

Overseas universities are also creating a smart learning environment through learner-led collaboration with standardized platforms and Web 2.0 learning tools. These changes, along with an

interest in developing and utilizing educational open sources for joint use of content, are currently being applied to a series of platforms and online learning tools that are being used by more than 40 million people in 213 countries and more than 70 universities around the world. In addition, the product line that occupies the LMS (Learning Management System) market consists largely of Blackboard, Sakai, Desire2Learn, and eCollege. It also intends to apply a class model derived for university students and verify its effectiveness.

As such, specific research issues to achieve research purposes are as follows.

(1)   First, which of the functions of a Moodle-based online learning system enhances creative collaboration?
(2)   Second, what components can enhance creative collaboration in a Moodle-based online class?
(3)   Third, is the Moodle-based class model effective in promoting creative collaboration among college students?

## 2. Literature Reviews

### 2.1. Creative Collaboration

Collaboration is a temporary relationship created to achieve a specific purpose among professionals, which is mainly used to share a workload that cannot be dealt with alone, thereby increasing productivity or reliably solving difficult problems. During this collaboration, creative collaboration aims to achieve creative, novel, and relevant results through shared creations developed by the contractors. It is known that creative collaboration affects creative outcomes through psychological mechanisms such as selection-in-action among cooperatives, production of many alternatives, reliability of alternatives through complementarity, and stability through interdependency involving interaction between the cooperators.

Beckman, Scott, and Wymore (2018) argue in their research that collaboration provides a variety of experiences for students and balances openness and resistance to change [6]. Collaboration also emphasizes the ability to interconnect the expertise of each student. As such, collaboration can be seen to contribute to providing students with a variety of experiences and interconnecting their forms of expertise. However, students do not have many opportunities to engage in highly collaborative activities at school. In particular, for college students, collaborative activities that require a lot of time due to major learning objectives are not frequently used as teaching methods. As online learning is being strengthened, it is necessary to find ways to enhance collaboration activities through online learning.

Edmondson (2012) discussed ways to develop team building skills among students: In Collaborative Innovation, students toggled regularly between inner and other forms of focus as a way to learn about both themselves and their peers. In part, this was facilitated by an explicit focus by faculty on developing teamwork [12]. In this way, various methods should be tested to promote interaction and collaboration among students rather than the instructional methods of professors.

In an online multi-user virtual environment, cooperatives can access a collaborative environment without restrictions of time and place, which can help students share different perspectives, or participate in alternative decisions and appropriate dialogue [13]. Therefore, an online learning system should be established in which students can not only learn knowledge and experience from knowledge-oriented professor-learning perspectives, but also create new knowledge and information based on knowledge information gained from the knowledge base.

Based on this theoretical review of creative collaboration, Creative collaboration involves the use of psychological mechanisms such as reflection among cooperators, the production of many alternative options, the reliability of alternatives developed through mutual complementation, and stability provided through interactions between cooperators.

*2.2. Moodle-Based Online Learning System*

Moodle (Modular Object-Oriented Dynamic Learning Environment) is an open source LMS developed in Australia in 2002 and has been available since the 1.0 version was released in March 2011. In a cooperative learning environment that uses Moodle, anyone can become a professor or a learner. Learners can participate in collaborative learning activities such as forums, Wiki, vocabulary, databases, and messages using Moodle. These learners' collaborative learning activities also improve the experiences of other learners. In particular, Moodle provides a number of ways to express and share the knowledge that learners possess. Moodle also provides a good tool for monitoring community learning activities. Through these collaborative activities, learners can create new knowledge on their own and further develop critical thinking and flexible creative problem-solving skills through mutual cooperation between members.

In this study, to determine the components of the Moodle Online Learning System, a prior study was analyzed as follows. First of all, discussion activities, such as forums, that are available as collaborative learning functions of the Moodle-based online learning system are effective and efficient in demonstrating creativity by facilitating the active participation of learners and providing them with new knowledge. In addition, the Wiki activity that organizes learning from discussion activities into writing and enables learner interactions with other learners increases creativity [14].

Real-time interaction is one of the advantages of an online learning system. In online learning, real-time interactions can be active to enable creative collaboration between professors and students rather than just uploading and managing of bulletin boards and learning materials. Message functionality also provides learners with the ability to freely express opinions about the content of classes and other opinions related to learning. These Moodle-based online learning systems are based on constructivist learning theories, enabling learner-oriented classes with the expectation of increasing the creative collaboration function among multiple learners.

Moodle can be modularized among the class members, and has features that facilitate free discussion and evaluation. In particular, Moodle has several functions that help with cooperative learning, and these can be actively changed, depending on the level or purpose of the learner. Because of these advantages, it has the largest number of institutions and users in the world, based on the open-source learning system area. In addition, the flexibility of cooperative learning is a key element in the recent eLearning trend that enables collaborative learning, breaking away from the existing eLearning forms where simple questions or tasks are presented. That is, the system flexibility of the various learning processes, such as collecting individual opinions, writing documents, and meetings, is becoming a very important element in the eLearning learning model.

The Moodle-related educational theory aspect is designed to promote collaborative learning on the basis of constructivism [4]. These examples are effectively shown in Moodle's online learning activity menu and have the ability to post replies on bulletin boards and evaluate and recommend comments from learners to each other. In the Mutual Assessment Menu, evaluations of student interdisciplinary tasks or participation in learning activities can be made available to enhance the creative collaboration function.

Furthermore, the Moodle learning system's blogs and Wiki make it easier for learners to create Web content and facilitates the creation of a cyber community. Using blogs and Wiki in education can facilitate learners' self-control and active learning. Blogging enables individuals to create a global cyber community by posting and sharing their interests, writings and links with others. In addition, the forum menu also enables active learning. The Beyer (2012) study found that Wiki activity can support learners' meta-cognition by asking questions to facilitate reflections on learning processes and outcomes [15].

*2.3. Development Environment and Procedure for Installing of the Moodle Online Learning System*

In this study, the hardware environmental central processing unit (CPU) specification of the Moodle online learning system developed to enhance creative collaboration is Intel Pentium Dual 1.73 GHz and

the periodic memory specification is 896 MB. With the software environment of the system, the operating system is Windows 7, the database is MySql, the web server is Apache, and the web technology is PHP. In addition, if you are installing Moodle on a Windows server, a Visual C++ redistributable file for Visual Studio 2012 must be installed at http://www.microsoft.com/en-us/download/details.aspx?id=30679 Visual C++ (x86 or x64). The disk space on the hardware requires a minimum capacity of 5 GB to store 200 MB of Moodle code and content. Processors need 1 GHz (min), 2 GHz dual-core or higher, and 512 MB (min) or more than 1 GB of memory is recommended. These Moodle installation requirements may vary depending on the specific hardware and software combination, type of use, and type of load, and additional resources may be required for the more frequently used sites.

The procedure for installing the Moodle in these Windows operating systems is as follows: First, download and unzip the Moodle version from https://download.moodle.org/. Second, run the Moodle installation file to register the service. Third, create mysql DB and grant the user authority. Fourth, the installation begins when you enter the Moodle server address into the web address in an internet browser. Fifth, launch an internet browser and select http://127.0.0.1 by running an installed file to activate the Moodle site in the local environment.

## 3. Research Methodology

This study develops a Moodle-based online learning system class model to enhance creative collaboration capabilities, and the following are the methods of study that were used. First, as a planning step, the functional elements of the online learning system used to enhance creative collaboration were analyzed based on the function of collaborative learning among the functions of the Moodle Online Learning System. Second, as a design step, the Moodle Online Learning System was designed by analyzing the link between the Moodle Online Learning System function and the creative collaboration function. Third, as a development stage, an online learning system class model was derived to enhance creative collaboration by applying the constructivist teaching–learning principle related to Moodle online learning. Third, after applying the developed Moodle-based class model to university students, the results were analyzed and shown to be effective in creative collaboration. The research procedure is shown in Figure 1.

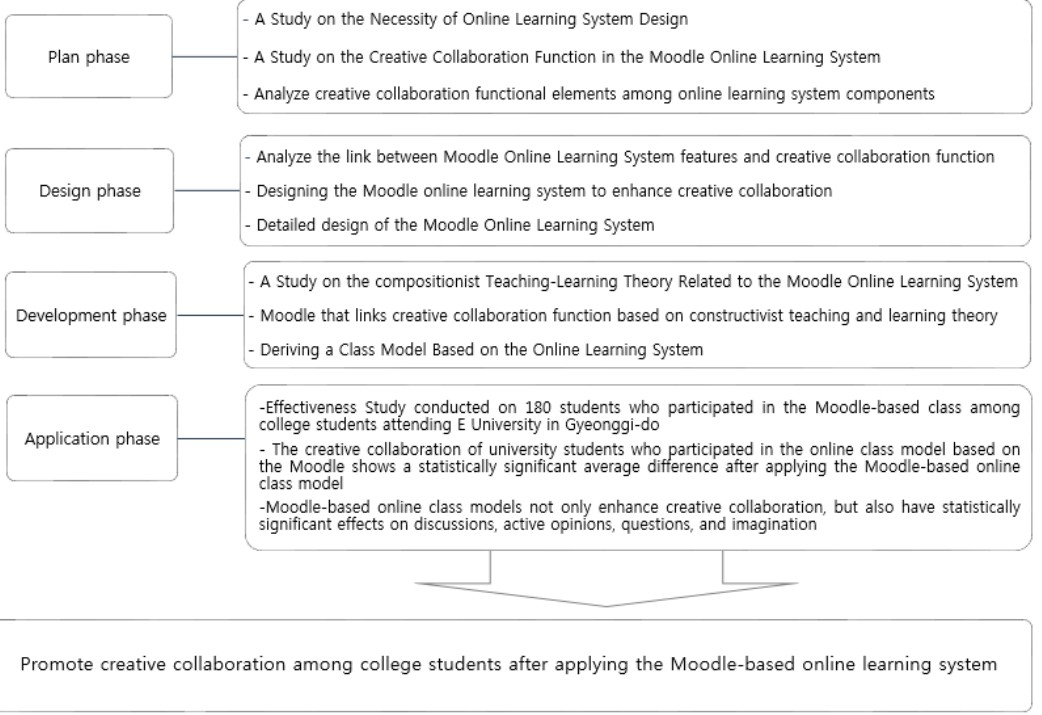

**Figure 1.** Research methods and procedures.

## 4. Results

### 4.1. Moodle Online Learning System Function and Creative Collaboration Function

Analysis of the functions of the Moodle online learning system that are highly relevant to the function of creative collaboration showed that the Moodle online learning system functions include Forum, Real-time convergence, Reciprocal evaluation, Wiki, and Blog, as shown in Table 1.

**Table 1.** Moodle's functions and creative collaboration functions.

| Moodle's Function | Creative Collaboration Function |
|---|---|
| Forum | -Online discussions among members on simple topics<br>-Discussions by question, topic, learners and groupings |
| Real-time conversation | -Brainstorming through Idea Generation Activity<br>-Online discussion of learners' assignments<br>-Exchange opinions on project progress by allowing learners to ask questions and answer them in real time |
| Reciprocal evaluation | -Function of Mutual Evaluation between learners |
| Wiki | -Create documents through cooperative work, create and paste maps<br>-Collecting documents shared by members is a possibility for members |
| Blog | -Interaction between members as a means of displaying their presence in a personal learning space or web can work with social network systems or e-portfolios<br>-Publish focused learning on personal interests and knowledge gained from learning to share with members |

Rifkin (2009) stated that classes should be transformed from centralized and top-down characteristics to having mutually democratic and networked characteristics, and that students should realize that they are also responsible for the education of the other students and have creative collaboration capabilities [16]. In regards to Moodle's functions, the forum is used to coordinate one's ideas with others through discussions among its members on a topic, and for members to have an open attitude to views that are different from their own. Real-time conversations in the second Moodle function enable brainstorming through idea-generating activities. They also enable a real-time exchange of views on project progress, thereby enabling creative collaboration among students.

The third Moodle function, mutual evaluation between learners, involves students learning to accept criticism from others and working to help each other. In the fourth Moodle function, a Wiki creates a sense of responsibility for learning outcomes by enabling students to write documents through creative collaboration. As Moodle's fifth function, blogs are a way for students to express their presence on the Web or in a private learning space, and to interact with other blog members to foster empathy through creative collaboration.

### 4.2. Development of the Moodle Online Learning System

The screen composition of the developed Moodle Online Learning System after the initial screen can be divided into five options, as shown in Figure 2. These options include My course, Site Home, Calendar, Personal file, and My Homepage. The sub-structure for My course includes Survey, Task, Chat Room, Vocabulary note, Wiki, Quiz, Forum, and Feedback to enable creative collaboration. A photo of the homepage of the developed Moodle-based online learning system is presented in the Appendix A.

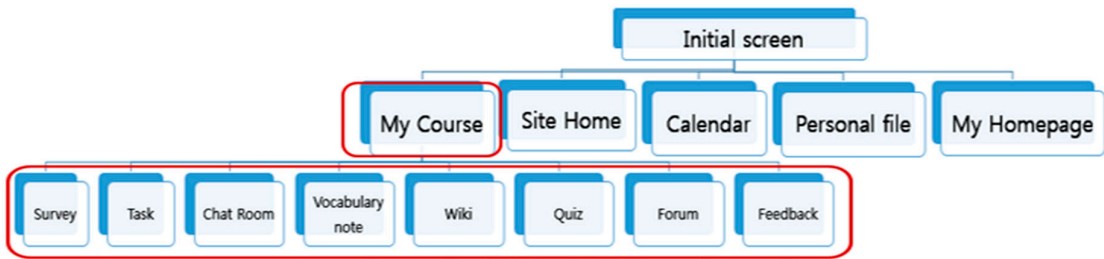

**Figure 2.** Basic structure of a system.

The web design storyboard is shown in Figure 3.

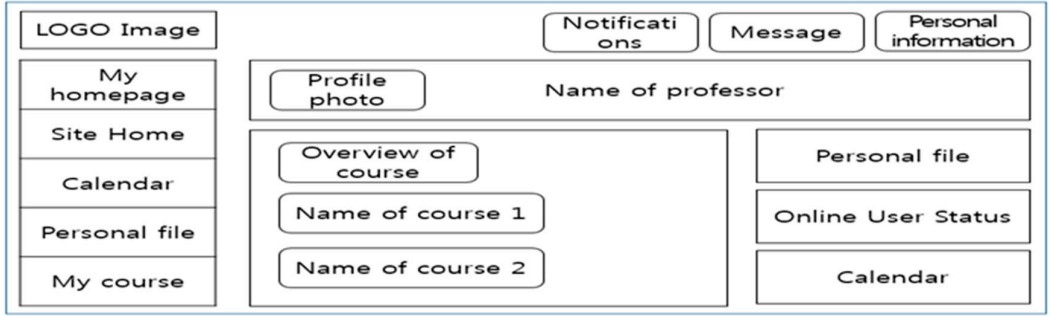

**Figure 3.** Web Design Story Board of a system.

*4.3. Class Model*

A systematic class model should be designed to effectively apply the Moodle-based online learning system to collaborative forms of education. It should also be developed to enhance learning convenience and accessibility for learners, and also to enhance the creative collaboration function of learners. To this end, the design of a class model using the Moodle-based online learning system was selected and organized into five categories using the organizing teaching–learning principle. These five principles include the composition process, learner-led learning, cooperative learning, practical tasks, and the role of a professor. The Moodle-based online class model using the constructivist teaching–learning principle is presented in Table 2.

**Table 2.** A Moodle-based Online Class Model Using Compositive Teaching and Learning Principles.

| Constructivist Teaching-Learning Features | Moodle-Based Online Class Model | Moodle's Creative Collaboration Function |
|---|---|---|
| Self-reflection | Problem-solving Learning and Evaluation | Mutual evaluation, forum |
| Learner-driven learning | Post Learner-led Ideas | Blog |
| Cooperative learning | Troubleshooting through learning-related conversations, discussions, and chatting | Forum, Wiki |
| Practical task | Presenting practical activities and tasks related to learning content | Quiz, Real Time Convers |
| The role of a professor | Helping and providing information as a guide | Real-time conversation |

Moodle's creative collaboration capabilities include the forum as a reflective learning element. It can be seen as one of the constructivist teaching–learning features in Table 2. The learner-driven learning element is a blog. The cooperative learning elements include forums and the Wiki. The real challenge factor is the quiz. Finally, the role element of the professor is real-time conversation.

As such, the Moodle-based online class model provides learners with more experience and interaction than is offered in a typical classroom environment. In particular, in the general classroom environment, learners will be able to experience a variety of interactions with other

users accessing the community, compared to limited space and interactions with people experienced in a classroom environment.

*4.4. The Results of a Creative Collaboration Analysis Based on the Application of a Moodle-Based Class Model*

Effectiveness was analyzed around college students attending four-year universities to find out the changes in creative collaboration following the application of the Moodle-based class model. First, the sample involved in the effectiveness study was 200 college students taking psychology classes, and with the exception of the high number of missing cases, the number of samples used in the analysis was 175. The general characteristics of the samples are shown in Table 3.

**Table 3.** General characteristics of research participants (N = 175).

| Variable | Group | Frequency (*n*) | Percent (%) |
|---|---|---|---|
| Sex | Male | 36 | 20.2 |
| | Female | 142 | 79.8 |
| Grade | 1 Grade | 4 | 2.2 |
| | 2 Grade | 35 | 19.7 |
| | 3 Grade | 82 | 46.1 |
| | 4 Grade | 57 | 32.0 |
| College | Department of infant education | 42 | 23.6 |
| | Department of optometrics | 8 | 4.5 |
| | Department of Food and Nutrition | 33 | 18.5 |
| | Department of Medical IT | 32 | 18.0 |
| | Department of Nursing Science | 52 | 29.2 |
| | Department of Medical Engineering | 2 | 1.1 |
| | Department of Dental Hygiene | 5 | 2.8 |
| | Department of clinical pathology | 1 | 0.6 |
| | Department of Medical Public Information Design | 1 | 0.6 |
| | Department of Medical Management | 1 | 0.6 |
| | department of emergency rescue | 1 | 0.6 |

In addition, SPSS Statistics version 23.0 was used for statistical analysis of the collected data. First, frequency analysis was used to identify the general characteristics of a variable. In addition, the two dependent sample *t* tests were used to compare pre- and post-creative collaboration in order to find out the effects of the Moodle-based class model application. The results are shown in Table 4.

**Table 4.** Results of two dependent samples for the change of creative collaboration with the application of a Moodle-based class model (N = 175).

| | Pre-Creative Collaboration | Post-Creative Collaboration |
|---|---|---|
| Mean | 3.18 | 3.46 |
| standard deviation | 0.92 | 1.05 |
| the number of cases | 175 | 175 |
| *t* | −4.18 *** | |
| *p* | 0.000 | |

*** $p < 0.001$.

The average of pre-creative collaboration is 3.18, and the standard deviation is 0.921; the average of post-creative collaboration is 3.46 and the standard deviation is 1.055. The results of the statistical significance of the difference between pre- and post-creative collaboration showed that t-statistics were −4.188 and significance probabilities were 0.000, which differed in the pre- and post-script scores of university students' creative collaboration by applying a Moodle-based class at a significant level of 0.000.

In addition, the Moodle-based online class model has been shown to have a positive effect on promoting creative collaboration among college students, as well as on discussion activities, active opinions, active questions and increased imagination. The result is as shown in Table 5.

**Table 5.** *t*-test results of two dependent samples for learning activities of university students by applying a Moodle-based class model (N = 175).

|  | Pre-Mean (SD) | Post-Mean (SD) | *t* | *p* |
|---|---|---|---|---|
| Discussion activity | 2.92 (0.891) | 3.15 (1.031) | −4.138 *** | 0.000 |
| Active expression of opinion | 3.13 (0.839) | 3.38 (0.978) | −4.759 *** | 0.000 |
| Active questions | 3.22 (0.992) | 3.32 (1.064) | −2.130 * | 0.035 |
| Imagination | 2.93 (0.901) | 3.09 (1.019) | −3.055 * | 0.003 |

*** $p < 0.001$, * $p < 0.05$.

## 5. Conclusions

In this study, a Moodle-based online learning system was developed and a class model was derived to enhance learners' creative collaboration capabilities. In addition, the results of creative collaboration were analyzed after applying the derived class model to university students. The conclusions drawn from these findings are as follows:

First, the development of the Moodle online learning system focused on the functions of forms, real-time conversation, reciprocal evaluation, Wiki, and blogs to promote creative collaboration among college students.

Second, the class model for promoting creative collaboration based on a Moodle-based online learning system consisted of self-reflection, learner-driven learning, cooperative learning, practical tasks, and the role of the professor.

Third, after verifying the effect of application of the Moodle-based class model for university students, a positive effect was found with regard to the creative collaboration of university students. In addition, the application of the Moodle-based online class model for college students had a positive effect on improving the discussion activities, active options, active questions, and imagination, in addition to creative collaboration.

This system can provide an opportunity to improve individual decision making, problem solving, criticism, and thinking skills in solving problems by providing learning individualization and a self-directed learning environment for learners, as well as the ability to quickly explore and share the latest, and a diverse range of information that is characteristic of the web. In addition, the lack of class time and lack of mutual cooperation learning due to a large number of students can be substantially supplemented through chat rooms, bulletin boards and e-mail activities, and active and diverse experiences can be learned through asynchronous interaction with learners, fellow learners, professors, experts, and by providing diverse multimedia resources indefinitely from the traditional teaching methods.

The implications of this study are as follows: First, in order to enhance interactivity among members and increase their creative collaboration capabilities, a class model based on an online learning system should be developed. Second, in order to promote creative collaboration, learners should use online learning systems to obtain a variety of learning opportunities. Third, the outputs obtained after cooperative learning using the online learning system should be accumulated and used for performance evaluation and course evaluation at a later stage.

In order for education to be effectively conducted using the Moodle-based online learning system developed in this study to promote creative collaboration, the creative collaboration function of the developed Moodle should be able to maximize the educational effect. To do this, online learning system-based, creative, collaborative teaching methods should be applied to online teaching design. In an online learning environment, learners can take the lead in finding a variety of information, interacting with other learners in real time and non-real time, and reconfiguring their own knowledge

and expression [17,18]. Furthermore, Ku Yang-mi et al. (2006) argue that the online learning environment helps to promote divergent thinking by giving learners the opportunity to constantly expand their thinking while freely accessing unexpected information [19]. The online learning environment is gaining traction as a learning environment suitable for creative problem solving and collaboration, helping to generate diverse and unique ideas that can be accessed across time and space, uploaded in one's mind, searched, reviewed, and communicated to others [20].

However, the online learning system developed in this study was implemented by customizing and implementing packages, which may have limitations in program modification. The online learning system developed in this study is an open-source system and may be limited in the use of some functions. This study is also to develop a Moodle-based online learning system class model that can enhance creative collaboration. Further research should be conducted to accurately identify the effectiveness of education in the application of the developed class model.

**Author Contributions:** Conceptualization, methodology, and Writing—Original Draft Preparation, E.K.; Writing—Review and Editing, H.P. and J.J.

**Funding:** This work was supported by the Ministry of Education of the Republic of Korea and the National Research Foundation of Korea (NRF-2018S1A5A8026816).

**Conflicts of Interest:** The authors declare no conflict of interest.

## Appendix A

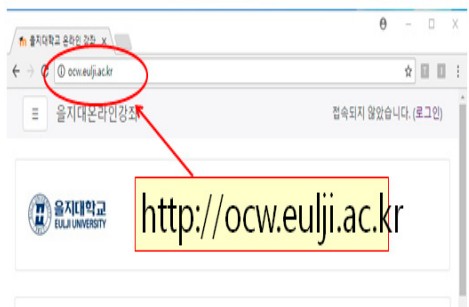

**(a)** This picture is shown as the result of inputting homepage address of developed Moodle system.

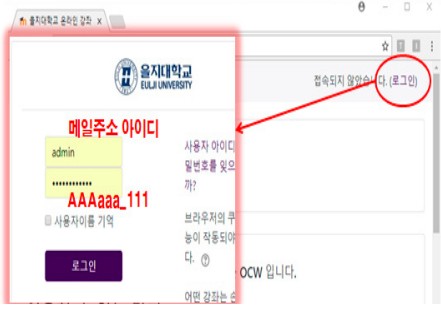

**(b)** This picture is the screen for logging into Moodle course.

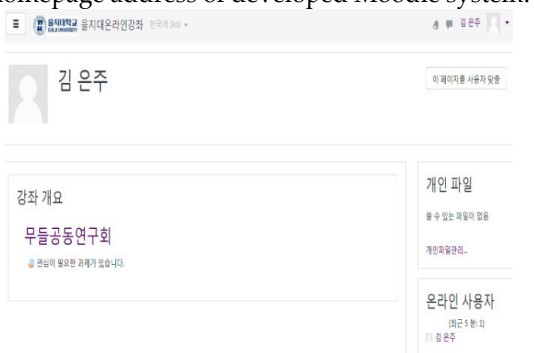

**(c)** This picture shows the subjects offered at Moodle system.

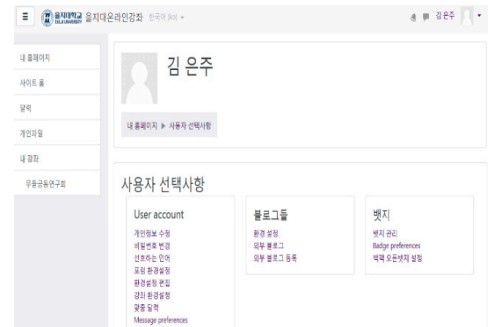

**(d)** The picture shows the personal homepage of the administrator (professor) who manages Moodle lecture.

**Figure A1.** *Cont.*

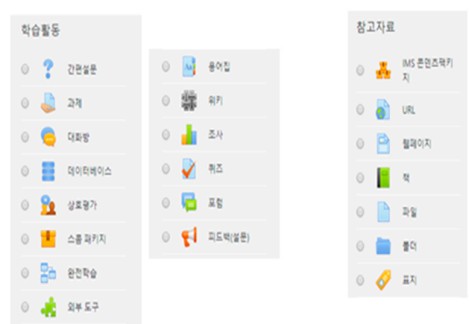 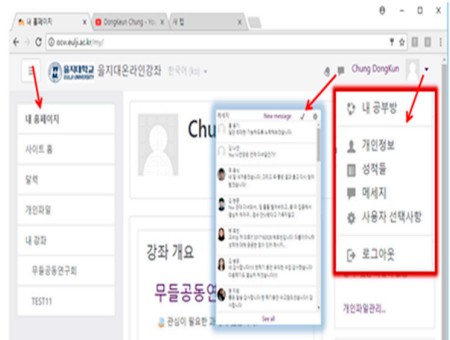

**(e)** This picture shows the type of activity included in My Course.

**(f)** This picture shows the process of active communication among students.

**Figure A1.** Photos of the homepage of the developed Moodle-based online learning system.

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
