# Peer review of "Development of a Class Model for Improving Creative Collaboration Based on the Online Learning System (Moodle) in Korea"

_2199-8531, doi:10.3390/joitmc5030067_

Round 1
Reviewer 1 Report
Dear authors, I appreciate that you have developed the part dealing with collaboration.
The text is still rather informative than research-based. Should the authors have experience with Moodle, learners feedback on how beneficial the tools were for developing the cooperation, comparison of face-to-face teaching and virtual cooperation (sustainability and permanence of the “utterances” in e.g. fora) they should use it to support the theoretical ideas they present).
Guessing, the authors have students’ feedback they may set the qualitative research design and analyse the data (in case they have data) what would be the basis for a research article.
Line 99 – the in-text citation of the source (15) Beckman, Sara, et al. “Collaborative Innovation: Exploring the Intersections among Theater, Art and Business in the Classroom.” Journal of Open Innovation: Technology, Market, and Complexity, vol. 4, no. 4, 2018, p. 52., doi:10.3390/joitmc4040052.
should be: Beckman, Scott, & Wymore (2018)
lines 165-182 – not important for the present study (it is relatively detailed information on technical equipment and as such is important information rather for technicians that for the Moodle users
Author Response
C1. Line 99 – the in-text citation of the source (15) Beckman, Sara, et al. “Collaborative Innovation: Exploring the Intersections among Theater, Art and Business in the Classroom.” Journal of Open Innovation: Technology, Market, and Complexity, vol. 4, no. 4, 2018, p. 52., doi:10.3390/joitmc4040052.
should be: Beckman, Scott, & Wymore (2018)
->Your opinion was of great help in revising my manuscript. I modified it according to the opinion you gave me. (Line 107)
C2. lines 165-182 – not important for the present study (it is relatively detailed information on technical equipment and as such is important information rather for technicians that for the Moodle users
->Your opinion is a very good one. But in this study, I didn't delete it because I hope it could help other researchers develop open-source Moodle systems. I am truly sorry for not being able to actively accept your opinion.

Reviewer 2 Report
The content is well presented and structured. The article can be published in the current form.
Author Response
C1. The content is well presented and structured. The article can be published in the current form.
->
Thank you very much for your positive opinion. Despite your positive comments, we have revised the content to reflect the opinions of other reviewers.
In particular, the following contents were added: After applying the Moodle-based class model as you advised, we analyzed the impact on the learner's creative collaboration. The results are described in 4.4.(The whole manuscript)

Reviewer 3 Report
There is a considerable revision on the contents, but yet still explicit explanation on the creative collaboration of students. Please add the part that how the students changed creatively and experienced collaboration in class.
That explanation is a key part of this paper, so you need to more explanation on the creative collaboration. As the researcher stressed, I think that explanation is needed to show that the creative collaboration of learners has been enhanced.
Author Response
C1. There is a considerable revision on the contents, but yet still explicit explanation on the creative collaboration of students. Please add the part that how the students changed creatively and experienced collaboration in class.
That explanation is a key part of this paper, so you need to more explanation on the creative collaboration. As the researcher stressed, I think that explanation is needed to show that the creative collaboration of learners has been enhanced.
-> Your advice was a big help. After applying the Moodle-based class model as you advised, we analyzed the impact on the learner's creative collaboration. The results are described in 4.4.(Line 256~272 and more modifications)

Round 2
Reviewer 1 Report
the scientific soundness increased I suggest authors add information on the sample (we know that it was relatively big sample n=175) and missing information on how the data were measuredAuthor Response
Comment : the scientific soundness increased I suggest authors add information on the sample (we know that it was relatively big sample n=175) and missing information on how the data were measured
Corrections(Line 261-270) : Effectiveness was analyzed around college students attending four-year universities to find out the changes in creative collaboration following the application of the Moodle-based class model. First, the sample involved in the effectiveness study was 200 college students taking psychology classes, and with the exception of the high number of missing cases, the number of samples used in the analysis was 175. The general characteristics of the samples are shown in Table 3.
Table 3. General characteristics of research participants (N=175)
Variable |
Group |
Frequency (n) |
Percent (%) |
Sex |
Male |
36 |
20.2 |
Female |
142 |
79.8 |
|
Grade |
1Grade |
4 |
2.2 |
2Grade |
35 |
19.7 |
|
3Grade |
82 |
46.1 |
|
4Grade |
57 |
32.0 |
|
College |
Department of infant education |
42 |
23.6 |
Department of optometrics |
8 |
4.5 |
|
Department of Food and Nutrition |
33 |
18.5 |
|
Department of Medical IT |
32 |
18.0 |
|
Department of Nursing Science |
52 |
29.2 |
|
Department of Medical Engineering |
2 |
1.1 |
|
Department of Dental Hygiene |
5 |
2.8 |
|
Department of clinical pathology |
1 |
.6 |
|
Department of Medical Public Information Design |
1 |
.6 |
|
Department of Medical Management |
1 |
.6 |
|
department of emergency rescue |
1 |
.6 |
In addition, SPSS Statistics version 23.0 was used for statistical analysis of the collected data. First, frequency analysis was used to identify the general characteristics of a variable. In addition, the two dependent sample t tests were used to compare pre- and post-creative collaboration in order to find out the effects of the Moodle-based class model application. The results are shown in Table 4.
Reviewer 3 Report
Overall, it has been revised in accordance with comments. Finally, there is something that must be added. It should be needed to add the explanation of the operational definition of creative collaboration and how to measure it in 4.4 part (relating Table 3~4 )
Author Response
Comment 1 : It should be needed to add the explanation of the operational definition of creative collaboration
Corrections 1(Line 126-129) : Based on this theoretical review of creative collaboration, Creative collaboration is the use of psychological mechanisms such as reflection among cooperators, production of many alternatives, reliability of alternatives through mutual complementation, and stability through interactions between cooperators
Comment 2 : how to measure it in 4.4 part (relating Table 3~4 )
Corrections 2(Line261-270) :
Effectiveness was analyzed around college students attending four-year universities to find out the changes in creative collaboration following the application of the Moodle-based class model. First, the sample involved in the effectiveness study was 200 college students taking psychology classes, and with the exception of the high number of missing cases, the number of samples used in the analysis was 175. The general characteristics of the samples are shown in Table 3.
Table 3. General characteristics of research participants (N=175)
Variable |
Group |
Frequency (n) |
Percent (%) |
Sex |
Male |
36 |
20.2 |
Female |
142 |
79.8 |
|
Grade |
1Grade |
4 |
2.2 |
2Grade |
35 |
19.7 |
|
3Grade |
82 |
46.1 |
|
4Grade |
57 |
32.0 |
|
College |
Department of infant education |
42 |
23.6 |
Department of optometrics |
8 |
4.5 |
|
Department of Food and Nutrition |
33 |
18.5 |
|
Department of Medical IT |
32 |
18.0 |
|
Department of Nursing Science |
52 |
29.2 |
|
Department of Medical Engineering |
2 |
1.1 |
|
Department of Dental Hygiene |
5 |
2.8 |
|
Department of clinical pathology |
1 |
.6 |
|
Department of Medical Public Information Design |
1 |
.6 |
|
Department of Medical Management |
1 |
.6 |
|
department of emergency rescue |
1 |
.6 |
In addition, SPSS Statistics version 23.0 was used for statistical analysis of the collected data. First, frequency analysis was used to identify the general characteristics of a variable. In addition, the two dependent sample t tests were used to compare pre- and post-creative collaboration in order to find out the effects of the Moodle-based class model application. The results are shown in Table 4.
This manuscript is a resubmission of an earlier submission. The following is a list of the peer review reports and author responses from that submission.
Round 1
Reviewer 1 Report
The authors need to clearly state in their paper what is new an original about developing a class using moodle and show results in comparison of what has done perviously. The reviewer previously used moodle for his own class but simply using moodel does not merit writing a manuscript for publication. Currently the manuscript is not proper format for publication since it does not build upon current knowledge in the field.
Reviewer 2 Report
I think this paper is important in educational research. But it needs some revision of contents.
first, the abstract and the introduction part should be modified to meet the journal standards. the abstract part is briefly described, not in accordance with the general journal standards.
second, it is necessary to describe the theoretical background and the process of system development which are derived from this study. There is an insufficient explanation as to why the constituent elements of the online learning system are constructed in this way.
third, a description of what the class model means and how it is constructed should be added. This paper seems to introduce the system simply, it gives the impression that the scientific analysis based on the thesis is lacking.
Fourth, the first and second aim of this paper There is no explanation for how it facilitated creative collaboration.
Totally, it is necessary to modify the form and content of this paper to fit the academic paper rather than simply constructing manual content that describes the online system.
Reviewer 3 Report
The use of the Moodle platform strengthens the academic community of the two participants in the educational process: teacher-student, creates a friendly learning environment, puts the student at the heart of the educational process, contributing to the achievement of educational goals, ie competencies, and by improving initial and continuous teacher training. But the lack of direct interactions can be considered a disadvantage of this platform. I think the article could be improved by comparing with similar research that is being done today.
Reviewer 4 Report
Dear authors,
I believe there are readers who might be interested in your text. However, there have been already published similiar text and I miss novelty in your text. I enumerate several problems I found readig your text. I hope this helps you focus on the reader's needs and expectations. I believe that adding your own experience (in the role of teacher or student) might be useful). With the aim to study Moodle possibilities to support learners' creativity and collaboration I would expect to find more facts about the tools, possibly learners' feedback, etc.
Your text needs to be proofread.
L 6 - The sentence is not clear (something missing..., „Results,...“)
Do you really mean results, or rather you state hypotheses?
L20 verb form
L24 – I would also stress that Moodle is the Learning Management System (LMS)
L25 – add ... ALONG with Blackboard (otherwise it sounds like BB was a part of Moodle)
is implemented – I would rather use – can be used on PC or mobile devices
L26 – cooperation learning – replace cooperative learning
L27 this expectations – incorrect grammar
L28 – develop – apply?
L47 professors-learning - professors-teaching?
The text needs a reorganisation of the ideas – there are passages where different ideas are mixed together – e.g 55 – 59
55 S1"In particular, Moodle provides a number of ways to
56 express and share the knowledge that learners have. S2 Moodle also provides a good tool for monitoring
57 community learning activities. S3 Through these collaborative activities, learners can create new
58 knowledge on their own and further develop critical thinking and flexible creative problem-solving
59 skills through mutual cooperation among members."
S1 – focus on tools
S2 – tracking students
S3 – students using activities created by the tools mentioned in S1
Idea in S2 is either not developed and clear or redundant – if you mean tracking students’ progress by teacher – it can be a separate paragraph, however, if you mean that the possibility to follow the changes by the students }e.g. in wiki) what can be seen as an advantage could be placed after the S2 and highlighted as a positive aspect, advantage of the tool
L69 These Moodle-based online learning systems – there refers to what? The Moodle is a system, which systems are mentioned in the sentence? Do you mean tools?
L73-75 Sentence structure
L 89 how does Sara Beckman et al fits to directly this context? If you want to point out to different studies focusing on different educational areas you have to mention more studies and more areas. Be careful, the next sentence is about the history of CL theory.
Lexical inappropriacy, e.g. L 96 - to develop learners' creative collaboration function (one does not develop function); L100 a tool for creative collaboration as a way to explore and solve educational methods (we do not solve the methods in this context)
In teaching-oriented literature, the term teacher-centered is used as an opposite to learner centered (rather than professor-centered)
L120-121 – could you be more specific – what do these numbers refer to? (the source ([13] link) is not available and thus the reader cannot find info)
In the study aim statement I recommend to omit - which is considered relatively good among open-source LMS among Moodle, Sakai, dotLRN, ILIAS, ATutor, Joomla, OLAT, and Claroline, (the comparison of LMSs can be a part of the literature review, but not necessarily the part of the study aim statement)
L128 – grammar revision needed
L134 – Figure 1 should define the research methods and procedure, however, research methods are not states but rather the phases of research are described
L138 – 146 – there is no Moodle analysis, but rather general description of its use in the world
Parts 4.2 and 4.3 do not relate to the aim of the study.
The whole text is rather descriptive than scientific. It might be interesting for people in case they are not familiar with LMSs, particularly Moodle, but there are a plethora of similar studies and from my point of view the novelty is missing.
Language needs serious proofreading as sometimes the meaning/understanding can be shifted because of e.g. incorrect sentence structure